# Accuracy of an Ultra-Wideband-Based Tracking System for Time–Motion Analysis in Tennis

**DOI:** 10.3390/s25041031

**Published:** 2025-02-09

**Authors:** Wenpu Yang, Jinzheng Wang, Zichen Zhao, Yixiong Cui

**Affiliations:** 1Sports Coaching College, Beijing Sport University, Beijing 100084, China; youthwp@bsu.edu.cn (W.Y.); zichenzhao@bsu.edu.cn (Z.Z.); 2School of Sports Engineering (China Sports Big Data Center), Beijing Sport University, Beijing 100084, China; wangjinzheng@bsu.edu.cn

**Keywords:** player-tracking technology, positional analysis, local positioning system, wearable devices, indoor tracking, physical analysis

## Abstract

Player-tracking systems provide vital time–motion and tactical data for analyzing athletic performance. Ultra-wideband (UWB) systems are promising for racquet sports due to their accuracy and cost-effectiveness compared to GNSS and optical systems. This study evaluated the accuracy of a UWB tracking system (GenGee Insait KS) for tennis-specific movements by comparing it with an optical motion capture system (VICON). Ten amateur players (International Tennis Numbers: 2–5) participated, performing seven exercises, including warm-up, agility drills, and tactical drills, with and without racquets. Raw data from both systems were processed to calculate the distances traversed. The average root mean square error between the two systems was 0.65 m (X-axis) and 0.76 m (Y-axis). Significant measurement discrepancies were observed (standardized mean difference: 0.86–1.95), except for jogging and walking exercises (*p* > 0.05). The overall percentage error was 16.29%. The intraclass correlation coefficient for distance measurements was 0.91, indicating good reliability. Tasks involving rapid acceleration and directional changes, such as the spider run, exhibited larger errors (mean bias: 4.13 m, effect size: 1.03). While the UWB system demonstrated acceptable accuracy for steady movements, it showed notable discrepancies during high-speed, tennis-specific activities. Overestimation due to arm movement and hip rotation suggests caution when applying arm-mounted UWB devices in training and competitive settings.

## 1. Introduction

Time–motion analysis is vital for evaluating elite tennis performance [1], providing detailed insights into movement patterns, external loads, and tactical execution effectiveness [2,3]. In a sport characterized by rapid directional changes and split-second decisions, understanding these dynamics is crucial for refining training and match strategies [4]. Accurately assessing acceleration, deceleration, and movement patterns is vital for injury prevention and performance enhancement [5].

Tracking athletes’ displacement and movement trajectory during strokes provides valuable insights into motion efficiency, allowing coaches to pinpoint specific areas for improvement. These data, which include detailed temporal and spatial information, offer actionable feedback that can directly inform adjustments to both technique and physical performance [6]. Player-tracking systems [7], including semi-automatic multiple-camera video systems (VID), radio-based local positioning systems (LPS), and global navigation satellite systems (GNSS), measure positions, velocities, and accelerations to quantify external loads [8,9,10]. System selection depends on accuracy, cost, and environmental applicability: GNSS excels outdoors, optical systems indoors, while LPS technologies like ultra-wideband (UWB) provide reliable indoor positioning with lower costs compared to optical setups [11].

UWB is particularly relevant for tennis due to its high spatial resolution, robustness, and ability to handle indoor environments with frequent net crossings and shadowed court areas. Unlike GPS (limited to outdoor use) or optical systems (sensitive to lighting conditions), UWB delivers centimeter-level accuracy by transmitting wideband signals (>500 MHz) [12], enabling precise tracking of rapid directional changes—a critical feature in tennis, where players average 4–6 direction shifts per rally [13]. Additionally, its high sampling rate (>25 Hz) and low latency (<200 ms) make it suitable for capturing high-speed movements such as accelerations and split-step reactions [14].

Despite its growing application in team sports [7], proper validation of UWB technology in tennis remains limited. Most studies employ standardized protocols [12] or speed-based references [15,16] that underestimate errors in tennis-specific maneuvers. Experimental data indicate that UWB’s average positioning error decreases from 14.2 cm to 11.3 cm when fused with inertial sensors during stroke execution [17]. However, challenges specific to tennis—such as arm-mounted sensor oscillations during serves/volleys and multipath interference near court boundaries—have yet to be addressed comprehensively [18]. Existing studies report 0.57–5.85% distance errors in controlled conditions [12], but acceleration errors exceed 33% during sharp directional changes [19]. This highlights the need for robust validation tailored to tennis.

Therefore, the purpose of the current study was to assess the accuracy of an arm-mounted UWB player-tracking system within tennis-specific indoor conditions. Validation of the UWB system was undertaken against a reference system (VICON) [13]. The comparison was carried out by conducting both common tennis movement tasks and simulated rally scenarios, such as test runs along predefined tracks, shuttle runs, and small court rallies. It was expected that the results would disclose the strengths and weakness of UWB in tracking external loads of tennis players during both match and training scenarios, particularly with regard to its accuracy across various movement types.

## 2. Materials and Methods

### 2.1. Participants

Ten amateur tennis players (with an International Tennis Number ranging from 2 to 5 and aged between 22 and 32) were recruited in the study after providing written informed consent. Prior to participation, all players received comprehensive verbal and written explanations of the study and did not present any physical limitations or musculoskeletal injuries that could potentially affect experimental process. This study was approved by the Research Ethics Committee of Beijing Sport University (Approval number: 2024265H), and all procedures were conducted in accordance with the Declaration of Helsinki.

The inclusion criteria were based on previous studies that investigated similar skill levels [11], with participants selected to have adequate tennis-specific movement abilities. The sample size of 10 participants was consistent with previous research in UWB-tracking studies, where sample sizes ranged from 4 to 13 participants. This sample size was deemed sufficient for this pilot study based on similar research and the nature of the experimental design.

### 2.2. Equipment and Venue

This study was conducted in a designated experimental area equipped with 16 VICON motion capture cameras (VICON Motion Systems Ltd., Oxford, OX5 1GB, UK) and 4 UWB anchors (GenGee Insait KS, Gengee Technology Co., Ltd., Shenzhen, China) for motion tracking (see Figure 1). The capture area was 8.8 × 7.8 m for VICON and 10 × 10 m for UWB. Boundaries were marked with adhesive tape to ensure participants remained within the measurable area.

The UWB system provided spatial–temporal data at 15 Hz and consisted of four anchors and wearable sensors attached to the participants’ left upper arm using a band. The VICON system, operating at 120 Hz, used infrared cameras and retro-reflective markers (10 mm diameter) placed at key anatomical landmarks to estimate the center of mass (CoM) based on a pelvic reconstruction method [20]. VICON is considered the gold standard for motion tracking due to its high accuracy (spatial error < 0.5 mm) [21].

For comparison, five retro-reflective markers were placed at key anatomical points (right shoulder, left shoulder, left anterior superior iliac spine, right anterior superior iliac spine, and sacrum) to estimate the CoM, following standard biomechanical tracking protocols [13,22]. The placement of markers aimed to provide a more biomechanically accurate reference than the UWB armband position for tracking whole-body motion.

### 2.3. Experiment Procedures

Once the setup was complete, participants were fitted with reflective markers and guided by the research group to perform the following tasks, which are categorized into three types: running tasks (without rackets), tennis-specific tasks (with rackets), and tennis practice (with balls). The detailed procedure for each task type is described below:Running Tasks (Without Rackets)
Circle Walk, Jog, Sprint and Slide Run: Participants performed three laps around a circular perimeter at different paces: walking, jogging, and sprinting. Participants also executed a slide run, which involves lateral movement maintaining a low center of gravity while effectively shifting weight from one leg to the other.Warm-Up Ladder: Participants completed various rope ladder drills, including two steps forward, two steps lateral shuffle, in and out, open–close, quick steps. The two steps forward drill focused on forward propulsion, while the two steps lateral shuffle emphasized lateral movement skills. The in and out drill required rapid foot placements inside and outside the ladder rungs to develop quickness, whereas the open–close drill concentrated on efficient foot positioning. Lastly, the quick steps drill aimed to maximize cadence and rhythm, promoting overall lower-body explosiveness.T-Run: Participants executed dynamic movements in a T-shaped area (4.5 m × 6 m), covering approximately 21 m in total, including forward sprint, lateral shuffle, and backward cross-step.Spider Run: Participants ran to five points arranged in a fan pattern (3 m distance) to retrieve balls, returning each to a central tray.
Tennis-Specific Tasks (With Rackets)
Tennis Tactics: Participants practiced various shots which mimic the actual tennis rally; participants start at serve, and then baseline FH and BH moving, FH or BH approaching, and then FH and BH volley; ends at smash.Girard Tennis Test [23]: Participants performed six directional movements (two forward sprints, two lateral, and two backward) each covering 3 m, totaling 36 m.
Tennis Practice (With Balls)
Simulation of Small-Court Rallying: A two-minute timed exercise simulating rallying.



Each activity was carefully monitored to ensure compliance with the prescribed movement patterns and to prevent participants from exceeding the designated boundaries. The tasks were selected based on their relevance to typical tennis movement patterns, aiming to replicate the dynamic and multi-directional movements required during actual matches. Running tasks like the T-run and spider run simulate fast sprints, lateral shuffles, and direction changes common in tennis, while the warm-up ladder drills focus on agility, footwork, and coordination, essential for both baseline and net play. Tennis-specific tasks, such as the tennis tactics and Girard tennis test, were chosen to simulate real tennis actions, including rapid directional changes and shot execution, further ensuring the tasks closely reflect on-court movements. The simulation of small-court rallying mimics the actual tennis rally scenario, helping participants engage in realistic movement patterns that are crucial for training and performance.

### 2.4. Data Processing

Due to disparate sampling frequencies (15 Hz for UWB and 120 Hz for VICON), both datasets were up-sampled to 360 Hz using the CubicSpline interpolation method [24]. This standardization of time intervals allows for more accurate comparison between the datasets, ensuring alignment in terms of sampling frequency. The CubicSpline method was chosen for its smooth, reliable results with minimal impact on data integrity. Future studies may consider conducting a sensitivity analysis to assess the effect of different interpolation methods on data accuracy.

Time alignment: This procedure begins by applying a Butterworth filter [25] to the input data, followed by filtering the processed data using the filter operation [25]. Subsequently, the filtered data undergoes normalization, scaling its range to achieve a mean of 0 and a standard deviation of 1. Then, the two sets of data undergo a sliding-window correlation analysis, extracting the segment corresponding to the optimal window from the original data segments.

Spatial alignment: Due to differences in the reference coordinate systems between the UWB device and the initial coordinate system of the VICON system, adjustments are made to align the two coordinate systems, ensuring they are within the same spatial scale. Procrustes analysis [26] is utilized to obtain the aligned target matrix. Subsequently, the aligned matrix undergoes reverse scaling and translation operations to restore it to the scale and position of the original data. Finally, data sequences are extracted from the scaled and translated matrix. All data processing and alignment procedures were implemented using Python 3.11 (Python Software Foundation, Wilmington, DE 19801, USA).

Processing of VICON Data: In this experiment, raw reflective marker data were collected from the VICON system. Missing portions of the data were supplemented using polynomial and relational interpolation methods. Polynomial gap filling was applied when trajectory data were available on both sides of the gap, as it relies on surrounding data points to interpolate the missing curves. All gap filling is performed in the professional optical-data-processing software QTM (Qualisys Track Manager, Version 2023.3 (Build 12577), RT Protocol versions 1.0—1.24 supported) [27].

### 2.5. Statistical Analysis

The accuracy of fundamental *XY*-position data was estimated using the root mean square error (*RMSE*) and percentage error. Lower *RMSE* values indicate smaller discrepancies. Errors of up to 10% are often deemed acceptable in field-based sports [28]. The *RMSE* and *percentage error* are calculated using the following formula:$$RMSE=\sqrt{\frac{1}{n}\sum_{i=1}^{n}\left(X_{i}-Y_{i}\right)}$$$$Percentage\;{Error}_{i}=\left(\frac{\left|X_{i}-Y_{i}\right|}{\left|X_{i}\right|}\right)\times 100$$
where $X_{i}$ represents the optical system measurements, $Y_{i}$ represents the UWB measurements, *i* represents each data point, and *n* is the sample size [29].

In addition to *RMSE*, the mean absolute error (MAE), max error, mean error, and standard deviation (SD) of error were calculated to estimate the distance of two system. The percentage error was also calculated to assess the relative accuracy of the UWB system compared to the optical system, using the formula:

This metric provides a normalized error measure that allows for direct comparison across different measurement scales.

Bland–Altman analysis was used to compare the measurements from the UWB and optical systems, assessing bias and limits of agreement. The Bland–Altman plot displayed the mean difference and 95% limits of agreement (±1.96 times the standard deviation) to evaluate the consistency between the two systems [30,31].

A linear mixed model was applied using Jamovi 2.3.28 (Jamovi Project, Sydney, NSW 2006, Australia) to compare differences in various movement routes and between the devices, with subjects treated as random effects to account for individual variability. The distances captured by the VICON and UWB systems were compared, and the *p*-values were reported. Additionally, Cohen’s d effect size (ES) was calculated along with its confidence interval (CI) to quantify the magnitude of the differences between the two systems [32]. Due to the small sample size and the non-normal distribution of the data, a Bootstrap [33] method was employed to estimate ES and its CI. This approach was used to avoid the normality assumption required by parametric methods. A total of 1000 resamples with replacement were generated from the original data for the calculation of the effect sizes and the estimation of the 95% CIs. Cohen’s d values were interpreted as follows: ES < 0.2 trivial, 0.2–0.6 small, 0.6–1.2 moderate, 1.2–2.0 large, and >2.0 very large [34].

Additionally, the intraclass correlation coefficient (ICC_(2,1)_) was estimated using SPSS 25 (IBM Corp., Armonk, NY 10504, USA), employing a two-way random effects model with absolute agreement to assess the reliability of UWB measurements across repeated trials. The ICC_(2,1)_ ranges from 0 to 1, with values indicating better reliability. An ICC_(2,1)_ > 0.75 indicates good reliability, while an ICC_(2,1)_ > 0.9 indicates excellent reliability [35,36].

## 3. Results

### 3.1. Position Accuracy

The RMSE for x and y coordinates between UWB and VICON averaged around 1 m. Specifically, the average RMSE values for the x and y coordinates were 0.65 m and 0.76 m, respectively, as shown by the dashed lines in Figure 2. These values reflect the overall accuracy of the position measurements between the two systems, with x and y discrepancies calculated from the distance differences shown in the box plots. Agility ladder (x = 0.31 m) and walk (x = 0.40 m) tasks had the smallest RMSE, while spider run (y = 0.90 m) and T-run (y = 1.07 m) had the largest.

Mean bias (Table 1) was lowest in circle movement drills (0.84–2.96 m) and increased for tennis-specific movements (0.99–4.13 m), with Girard and tactics exercises around 6 m. Rally practice exhibited the highest mean bias at 52 m. The overall percentage error was 16.29%. Walk, sprint, and T-run demonstrated acceptable percentage errors (<10%), while the percentage errors for other tasks remained below 30%, except for the practice task, which exhibited the poorest accuracy with a percentage error of 49%.

For circle walk and jog tasks, no significant differences were found between UWB and VICON distances; all other tasks showed significant differences (*p* < 0.01). Circle slide and warm-up ladder tasks had confidence intervals for effect sizes that did not include zero, indicating meaningful differences, while other tasks had overlapping intervals. The largest confidence interval for effect size was in the practice task (1.95) and the smallest in the warm-up task (0.86). Remaining tasks ranged from 0.97 to 1.54.

Figure 3 shows that UWB recorded higher distances than VICON in all complex running tasks, except for the simple linear walk, jog, and sprint tasks, where UWB measurements were lower than those of VICON.

### 3.2. Inter-Unit Reliability

In the Bland–Altman analysis (Figure 4), the UWB system generally underestimated distances compared to the VICON system, except in uniform and regular movement tasks (*circle run*, *jog*, and *sprint*), where mean differences exceeded zero. In other tasks, UWB measurements were shorter, with deviations exceeding 10 m only in the small-court practice task. The scatter point distribution indicated variability across tasks, though outliers were minimal, showing consistent differences overall.

The overall ICC_(2,1)_ was 0.913, suggesting strong agreement. However, measurement accuracy varied by exercise; agility ladder, T-run, and fixed tactics tasks showed lower reliability, with ICC_(2,1)_ from −0.28 to 0.14, while most differences remained within the 95% limits of agreement (LoA) (Figure 5).

## 4. Discussion

This study evaluated the accuracy of a UWB-based tracking system for time–motion analysis in tennis. The findings demonstrate that UWB technology offers a high level of precision, with an ICC_(2,1)_ of 0.913, indicating almost perfect agreement. This aligns with previous research highlighting UWB’s potential in sports tracking [37]. However, significant exercise-dependent variations in measurement accuracy were observed. While UWB positioning technology provides reliable measurements for certain exercises, its performance varies across different movement types, particularly during high-intensity or complex drills.

In this study, the overall RMSE for the y-axis was significantly higher than that for the x-axis following the coordinate system conversion between UWB and VICON systems. This discrepancy may be attributed to both movement patterns and the range of motion. Specifically, the testing scenarios involved more frequent movement along the y-axis, while movement along the x-axis was relatively limited, making the y-axis data more susceptible to noise and measurement errors, resulting in a higher RMSE. Furthermore, the greater range of motion in the y-axis likely contributed to an increase in cumulative errors, whereas the reduced movement in the x-axis may have helped mitigate error accumulation. These differences highlight the influence of movement characteristics on measurement errors [38].

The UWB-based tracking system was found to overestimate distances during complex tasks, likely due to multiple contributing factors. Signal reflection and multipath interference introduce measurement errors, particularly during rapid accelerations and directional changes, where shifts in body position complicate signal interpretation [16]. Additionally, the dynamic response limitations of UWB devices—such as mechanical strain and electronic interference during high accelerations—further impact accuracy. Insufficient update rates can also lead to delayed tracking of fast movements, exacerbating these measurement errors. While UWB demonstrates strong performance in basic movement patterns, its accuracy diminishes during complex, high-intensity tasks, especially those involving rapid direction changes and accelerations. This is consistent with findings in similar studies [13], such as those evaluating LPS and GPS technologies, where errors increased with higher speeds and more complex movements. For example, LPS demonstrated better positional accuracy than VID and GPS in tracking athletes, but accuracy declined during high-speed and multidirectional movements (>40% deviations from the reference system for each of the technologies), similar to our findings with UWB. To improve UWB’s tracking performance in these scenarios, future research should explore combining Inertial Measurement Units (IMUs) or optical motion capture with UWB, as IMUs provide real-time data on velocity and orientation changes, which are critical in fast, multidirectional tasks. Additionally, enhancing UWB’s update rates and minimizing environmental interferences could further improve its accuracy in dynamic environments.

Environmental factors, including electromagnetic interference and antenna placement, further impacted measurement quality. Additionally, manual segmentation of tasks introduced human error [37], and the fixed arm-mounted position of the UWB device contributed to discrepancies during dynamic movements, such as rapid direction changes. The natural swing of the arm and early hip rotation amplified these errors, particularly in tasks like spider run and tactics, where unpredictable movements are prevalent.

Variability in accuracy was particularly evident in exercises such as agility ladders, T-runs, and fixed tactics, with ICC values ranging from −0.28 to 0.14. Linear movements, such as jogging and walking, demonstrated acceptable accuracy, while complex, multidirectional movements presented significant challenges due to signal distortion and sensor placement. These discrepancies highlight the limitations of UWB systems in high-intensity tasks and underscore the importance of evaluating their performance in tennis-specific contexts.

UWB technology performed well in predictable, basic conditioning exercises, with mean biases ranging from 0.8 to 2.9 meters. However, in complex tennis-specific tasks, such as rally practice, the mean bias increased significantly (~52 meters), demonstrating UWB’s limitations in tracking dynamic and unpredictable movements. This calls for caution when applying UWB technology in tennis-specific training and match play.

While video analysis systems can provide the precision needed for tactical analysis, their high cost and impracticality underscore the potential of UWB as an alternative for regular use. However, due to its current limitations, UWB is more suitable for basic movement tracking than for precise tactical analyses.

Based on the results, the UWB system performed reliably in simpler, steady movements but showed reduced accuracy in more dynamic, fast-paced tennis actions, such as rapid direction changes. To enhance tennis performance, training must be sport-specific, as continuous linear running alone is insufficient to meet the demands of the game. The meta-analysis of tennis physical demands highlights that training should replicate the multifaceted physical demands of competition, including accelerations, decelerations, changes in direction, and high-speed movements [4]. Footwork drills should incorporate strength, agility, and speed to address these directional changes. Due to the limitations of UWB technology in accurately measuring acceleration and directional changes during complex movements, its application in tennis training and match play should be approached with caution. However, the overall error did not significantly vary across different movement patterns, and the UWB system demonstrated good inter-unit reliability. Therefore, while the UWB system is suitable for certain tennis applications, especially basic movement tracking, its limitations in high-speed, multi-directional movements should be considered when performing more precise tactical analyses.

## 5. Limitations and Future Research

The primary limitation of this study was ensuring that participants accurately followed the marked lines during the movement tasks. Although great care was taken to ensure precision, the effective range of the gold standard optical capture system proved to be inconsistent throughout the experiment. This variability in the system’s range, particularly during circular movements, led to fluctuations in the consistency of the distances covered across different trials. Additionally, the design of certain movement tasks allowed for a degree of autonomy and randomness in participants’ actions, which further contributed to variability. Consequently, noticeable differences in running distances were observed among different participants and between repeated trials, impacting the overall precision of the data.

Another limitation stemmed from the inherent complexity of real-world movement patterns in dynamic sports. While the optical capture system is highly accurate in controlled environments, its performance may degrade in real-world conditions, especially in sports like tennis, where rapid, multi-directional movements and quick accelerations occur. This limitation underscores the need for more robust validation of tracking systems in dynamic, real-game conditions.

Finally, this study was conducted solely in indoor settings, which may limit the generalizability of the findings to outdoor environments. Since tennis is a sport played both indoors and outdoors, the reliability and validity of the tracking systems in outdoor conditions remain unclear. Future research should aim to replicate these experiments in outdoor settings to assess the system’s performance across different environmental conditions. Additionally, a larger and more diverse sample of athletes should be included to ensure the robustness and applicability of the results across various playing styles and skill levels.

Future studies should also explore the integration of multiple tracking technologies, such as combining UWB with optical or inertial sensor systems, to improve tracking accuracy in dynamic sports. Additionally, there is a need for validation studies that examine not only the technical performance of these systems but also their practical impact on coaching and performance optimization in sports like tennis.

## 6. Conclusions

The study reveals that the tested UWB system offers acceptable accuracy in recording conventional linear and low-intensity acceleration movements. However, noticeable differences in the system from the criterion measures were observed when tennis-specific exercises involving high-intensity acceleration, deceleration, and changes in direction were considered. Therefore, the system should be applied with caution when tracking dynamic and multidirectional tennis movements. Meanwhile, the potential limitation of arm-mounted UWB sensors highlight the need for further algorithm refinement as well as the choice of sensor placement. Additionally, to enhance its reliability and applicability in tennis, future research should focus on evaluating UWB performance under real-world indoor and outdoor court conditions, as well as in scenarios with varying signal quality. Finally, it is essential to further validate IMU-derived load metrics specific to tennis strokes to ensure more accurate and comprehensive measurement of players’ technical and physical performance.

## Figures and Tables

**Figure 1 sensors-25-01031-f001:**
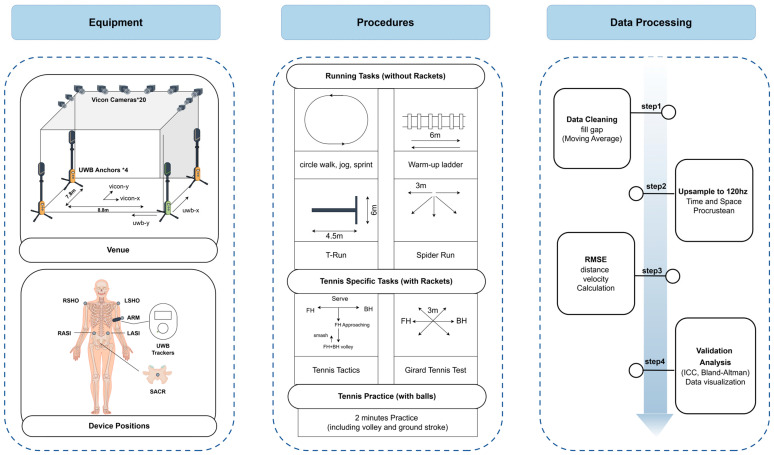
Experiment procedures.

**Figure 2 sensors-25-01031-f002:**
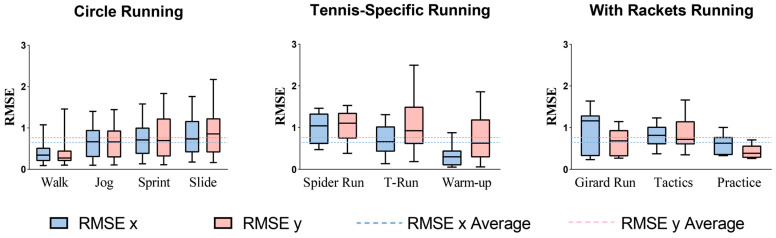
Root mean square errors (RMSE) for different tests: comparison of UWB and optical motion capture systems.

**Figure 3 sensors-25-01031-f003:**
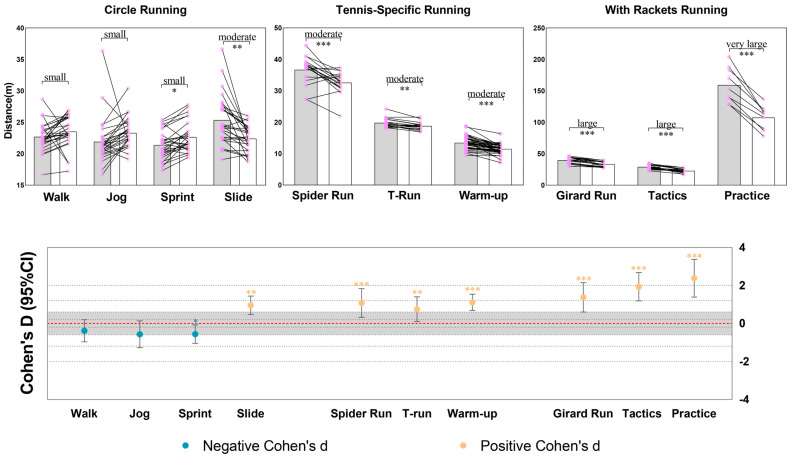
Comparison of distance measurements between VICON and UWB systems and standardized mean difference with 95% confidence interval. Note: The sign of Cohen’s d indicates the direction of the difference between the two groups in the variable being compared. A positive Cohen’s d suggests that the mean distance measured by UWB system is greater than that measured by VICON, whereas a negative value indicates that the distance measured by UWB system is smaller than that of VICON system.

**Figure 4 sensors-25-01031-f004:**
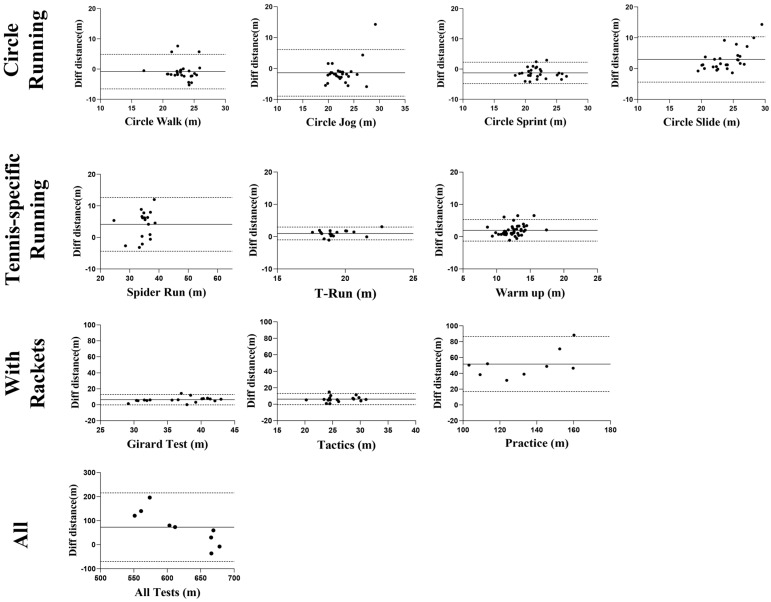
Bland–Altman plot comparing UWB and VICON distance measurements.

**Figure 5 sensors-25-01031-f005:**
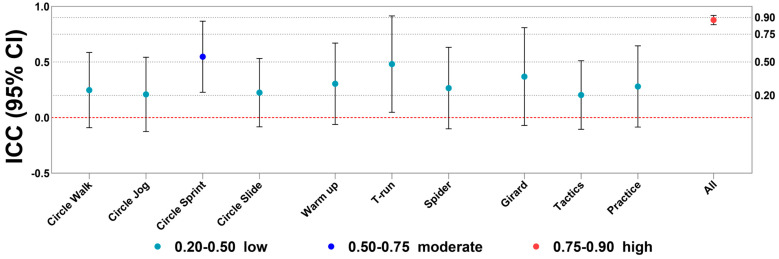
ICC_(2,1)_ for distance measured by UWB and VICON within each exercise.

**Table 1 sensors-25-01031-t001:** Mean differences (m) between two systems within different tests.

	N	RMSE (m)	MAE (m)	Max Error (m)	Mean Error (m)	SD of Error (m)	Average Error (%)	Error Interpretation
**Circle Running**								
Walk	27	2.98	2.28	7.62	−0.84	2.91	9.96	acceptable
Jog	27	4.02	3.02	14.28	−1.40	3.85	12.86	poor
Sprint	27	2.18	1.87	4.20	−1.27	1.80	8.18	acceptable
Slide	27	4.72	3.18	14.30	2.96	3.74	14.42	poor
**Tennis-Specific Running**								
Agility ladder 1: 2 steps forward	9	2.41	2.02	5.10	1.77	1.74	17.62	poor
Agility ladder 2: 2 steps sideways	9	2.37	1.52	6.08	1.52	1.93	16.42	poor
Agility ladder 3: Split steps	9	3.77	3.14	6.54	3.14	2.21	27.70	poor
Agility ladder 4: Shuffle in and out	9	2.36	2.23	3.41	2.225	0.85	20.27	poor
Agility ladder 5: Fast feet	9	1.45	1.27	2.61	1.14	0.95	10.93	poor
All agility ladder	45	2.58	2.03	6.54	1.96	1.70	18.59	poor
Spider Run	18	5.90	5.07	12.00	4.13	4.34	16.17	poor
T-run	18	1.38	1.19	3.07	0.99	1.00	6.34	acceptable
**With Rackets**								
Girard Run	18	7.07	6.31	14.28	6.31	3.27	19.23	poor
Tactics	18	7.14	6.29	14.81	6.29	3.47	29.04	poor
Practice	9	54.32	51.70	88.22	51.70	17.67	49.00	poor
**Summary**								
Total distance	9	100.07	82.57	196.16	−72.73	72.90	12.72	poor
All tests	234	11.46	5.02	88.22	3.66	10.88	16.29	poor

Note: N, number of trials; MAE, mean absolute error; SD, standard deviation; RMSE, root mean square error; Max error, maximum error; MAE and RMSE were determined with respect to the calculated VICON distance.

## Data Availability

The raw data supporting the conclusions of this article will be made available by the authors on request.

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
