# Peer review of "Accuracy of an Ultra-Wideband-Based Tracking System for Time–Motion Analysis in Tennis"

_sensors, 2025, doi:10.3390/s25041031_

Round 1
Reviewer 1 Report
Comments and Suggestions for Authors
This paper compares the positioning accuracy of UWB with respect to the optical system, various real experiments with different tennis sport motions are conducted, and the results are analyzed. There are some comments listed as follows:
1) The UWB tag was fixed on the participants' arm, so the obtained UWB coordinates cannot be recognized as the center of the participants. Additionally, the discrepancies may be significantly different in the various test tasks caused by the arm movements.
2) In the Introduction, more technical details about the positioning principle of the UWB could be introduced.
3) What are the meanings of the two abbreviations, SMD and ES, in the Abstract?
4) A possibly typo in Line 136? rungs -> run?
5) Different metrics were used to analyze the experimental results, e.g., Cohen's d effect size (ES), intraclass correlation coefficient (ICC), how to calculate these metrics?
6) The text in some figures are unclear.
7) Where is the Section 3.2?
8) The title of Section 3.3 is not appropriate.
Author Response
Reply to Reviewer #1
Dear reviewer,
Thank you very much for your time involved in reviewing the manuscript and your very encouraging and valuable comments on the merits. We have revised our manuscript accordingly.
Comments:
Open Review
Comments and Suggestions for Authors
This paper compares the positioning accuracy of UWB with respect to the optical system, various real experiments with different tennis sport motions are conducted, and the results are analyzed. There are some comments listed as follows:
Response:
To facilitate your review of this document, we will begin by retyping your comments in italic font, followed by our responses to those comments.
Comments 1:
The UWB tag was fixed on the participants' arm, so the obtained UWB coordinates cannot be recognized as the center of the participants. Additionally, the discrepancies may be significantly different in the various test tasks caused by the arm movements.
Response 1:
We agree with your concern regarding the discrepancies caused by the arm-mounted UWB tag, as the coordinates obtained may not represent the center of mass and could vary significantly due to arm movements. However, this system was specifically designed as an armband for practicality and minimal interference during training and competition. Consequently, the primary purpose of our study was to evaluate how this wearable configuration influences tracking accuracy and to determine the extent to which these errors impact its effectiveness in real-world training and match scenarios.
Comments 2:
In the Introduction, more technical details about the positioning principle of the UWB could be introduced.
Response 2:
Thank you for your valuable suggestion. In response to your comment, we have expanded the discussion on the positioning principle of UWB technology in the Introduction.
Comments 3:
What are the meanings of the two abbreviations, SMD and ES, in the Abstract?
Response 3:
We added the full names and removed all abbreviations.
Comments 4:
A possibly typo in Line 136? rungs -> run?
Response 4:
Thank you for pointing this out. The term "ladder rungs" is correct in this context, as it refers to the individual sections of the agility ladder used in the "In & Out" drill. The "rungs" are the horizontal parts of the ladder that participants must step in and out of. Therefore, no change is needed here, but we really appreciate your attention to the details.
Comments 5:
Different metrics were used to analyze the experimental results, e.g., Cohen's d effect size (ES), intraclass correlation coefficient (ICC), how to calculate these metrics?
Response 5:
We have clarified the calculation methods for Cohen's d effect size (ES) and Intraclass Correlation Coefficient (ICC) (using a two-way random effects model with absolute agreement) in the revised manuscript. We provided the necessary explanations for how they were applied to assess differences and reliability, respectively.
Comments 6:
The text in some figures are unclear.
Response 6:
The text in the figures has been made clearer by increasing the font size and ensuring better contrast for readability, which have been implemented in the updated manuscript.
Comments 7&8:
Where is the Section 3.2?
The title of Section 3.3 is not appropriate.
Response 7&8:
Thank you for your observation.
Section 3.2 was mistakenly omitted, and it has now been added to the manuscript. The title of Section 3.3 was incorrect and has been revised to match the correct title for that section. These errors have been corrected in the updated manuscript.
Reviewer 2 Report
Comments and Suggestions for Authors
The manuscript presents an interesting study on the application of Ultra-Wide-Band (UWB) technology for tracking tennis-specific movements. However, there are several areas where the study requires significant improvement to be published in Sensors.
1. Participant Details: The sample size of 10 amateur players is relatively small for drawing generalizable conclusions. The authors should elaborate on the rationale for this sample size and discuss its limitations in greater detail. Including participants with a wider range of skill levels (e.g., elite or professional players) would strengthen the findings.
2. Movement Tasks: The manuscript lists a variety of tasks, but the reasons for selecting specific tasks (e.g., agility ladders, T-runs) are not clearly justified. Were these tasks representative of typical tennis movement patterns? More justification is needed.
3. Data Processing: The manuscript mentions upsampling the data from both systems to 360 Hz. However, no justification is provided for this decision, nor is the impact of upsampling on data integrity discussed. Consider including sensitivity analysis to evaluate the effect of different interpolation methods.
4. Error Metrics: While RMSE is an appropriate metric, the manuscript would benefit from additional statistical methods to assess accuracy, such as percentage errors or consistency in specific movement contexts.
5. Task-Specific Results: The variability in results across tasks (e.g., RMSE differences between agility ladder and rally practice) is not thoroughly analyzed. Why did the UWB system perform better for linear movements and poorly for dynamic drills? A deeper dive into the underlying causes (e.g., signal interference, device placement) is necessary.
6. Comparison to Literature: The discussion lacks sufficient comparison to prior research. The authors should connect their findings to studies on UWB systems in other sports or tennis-specific tracking technologies (e.g., GNSS or optical systems).
Comments on the Quality of English LanguageThe manuscript's English expression is generally clear but requires simplification of overly technical terms, consistency in terminology and tenses, correction of minor grammatical errors, and refinement of long or complex sentences for better readability and accessibility.
Author Response
Reply to Reviewer #2
Dear reviewer,
Thank you very much for your time involved in reviewing the manuscript and your very encouraging and valuable comments on the merits. We have revised our manuscript accordingly.
To facilitate your review of this document, we will begin by retyping your comments in italic font, followed by our responses to those comments.
Comments :
Comments and Suggestions for Authors
The manuscript presents an interesting study on the application of Ultra-Wide-Band (UWB) technology for tracking tennis-specific movements. However, there are several areas where the study requires significant improvement to be published in Sensors.
Response :
Thank you for your constructive feedback. We have carefully addressed all the identified areas for improvement to enhance the quality and clarity of the manuscript.
Comments 1:
Participant Details: The sample size of 10 amateur players is relatively small for drawing generalizable conclusions. The authors should elaborate on the rationale for this sample size and discuss its limitations in greater detail. Including participants with a wider range of skill levels (e.g., elite or professional players) would strengthen the findings.
Response 1:
The sample size of 10 was selected based on a review of similar UWB tracking studies, where sample sizes ranged from 4 to 13 participants. Given the nature of this pilot study, we believe the sample size is sufficient for detecting differences in tracking accuracy. Additionally, compared to previous studies, this study includes a broader range of test tasks, which further supports the adequacy of the sample size. Future studies with larger sample sizes may enhance the generalizability of the findings.
Comments 2:
Movement Tasks: The manuscript lists a variety of tasks, but the reasons for selecting specific tasks (e.g., agility ladders, T-runs) are not clearly justified. Were these tasks representative of typical tennis movement patterns? More justification is needed.
Response 2:
Thank you for your comment. We selected these tasks based on their relevance to typical tennis movement patterns, ensuring that they closely mimic the types of movements players perform on the court. The Running Tasks such as the T-Run and Spider Run were chosen to simulate multi-directional, dynamic movements similar to those required during a match, such as sprints, lateral shuffles, and direction changes. The Warm-Up Ladder drills (e.g., In & Out, Open-Close) emphasize quick footwork, agility, and coordination, which are essential for tennis players during both baseline and net play. The Tennis-Specific Tasks (such as the Tennis Tactics and Girard Tennis Test) directly simulate real tennis situations, including movement between shots and rapid directional changes. These tasks aim to replicate the on-court demands of tennis, which require a combination of explosive speed, agility, and controlled footwork.
We have added this explanation to the manuscript to clarify the rationale behind task selection.
Comments 3:
Data Processing: The manuscript mentions upsampling the data from both systems to 360 Hz. However, no justification is provided for this decision, nor is the impact of upsampling on data integrity discussed. Consider including sensitivity analysis to evaluate the effect of different interpolation methods.
Response 3:
We have now provided a justification for upsampling both datasets to 360Hz. The decision was made to standardize the time intervals between the data points of both systems to enable more accurate comparisons. We acknowledged that while the CubicSpline interpolation method was used, which is commonly reliable for this purpose, we included a note that sensitivity analysis could be conducted in future studies to evaluate the impact of different interpolation methods on data accuracy. This would help assess any potential effect on data integrity and ensure the robustness of the results.
Comments 4:
Error Metrics: While RMSE is an appropriate metric, the manuscript would benefit from additional statistical methods to assess accuracy, such as percentage errors or consistency in specific movement contexts.
Response 4:
Thank you for your valuable suggestion. In response, we have added percentage errors as an additional metric to assess the accuracy of the UWB system compared to the optical system. The percentage error was calculated for each measurement, providing a normalized error measure that allows for direct comparison across different measurement scales. This addition enhances the accuracy assessment by providing a relative measure of error. The calculation of percentage errors has been included in the Statistical Analysis section of the manuscript, and the results are presented in Table 1 of the Results section.
We believe this addition strengthens the manuscript by providing a more comprehensive view of the measurement accuracy.
Comments 5:
Task-Specific Results: The variability in results across tasks (e.g., RMSE differences between agility ladder and rally practice) is not thoroughly analyzed. Why did the UWB system perform better for linear movements and poorly for dynamic drills? A deeper dive into the underlying causes (e.g., signal interference, device placement) is necessary.
Response 5:
Thank you for pointing this out. We have expanded the discussion to analyze the variability in results across tasks, providing detailed explanations for the UWB system's stronger performance in linear movements and its challenges with dynamic drills. We have attributed these discrepancies to factors such as signal reflection, multi-path interference, device placement, and the dynamic response limitations of the UWB system. Additionally, comparisons with findings from similar studies were included, and potential solutions such as integrating IMUs or improving update rates were suggested to address these issues.
Comments 6:
Comparison to Literature: The discussion lacks sufficient comparison to prior research. The authors should connect their findings to studies on UWB systems in other sports or tennis-specific tracking technologies (e.g., GNSS or optical systems).
Response 6:
Thank you for your suggestion. We have incorporated additional comparisons to prior research, highlighting how our findings align with studies on LPS and GPS technologies, particularly regarding accuracy decline during high-speed and multidirectional movements. We have also proposed future research directions to enhance UWB's performance, such as integrating IMUs or optical systems and improving update rates.
Comments 7:
Comments on the Quality of English Language
The manuscript's English expression is generally clear but requires simplification of overly technical terms, consistency in terminology and tenses, correction of minor grammatical errors, and refinement of long or complex sentences for better readability and accessibility.
Response 7:
Thank you for your feedback. We have carefully reviewed the manuscript to simplify overly technical terms, ensure consistency in terminology and tenses, correct grammatical errors, and refine complex sentences to enhance readability and accessibility.
Reviewer 3 Report
Comments and Suggestions for Authors
This study evaluated the accuracy of a UWB tracking system (GenGee Insait KS) for tennis-specific movements by comparing it with an optical motion capture system (VICON).Ten amateur players (ITN 2–5) participated, performing seven exercises, including warm-up, agility drills, and tactical drills, with and without racquets.The UWB system demonstrated acceptable accuracy for steady movements but exhibited significant discrepancies during dynamic tennis-specific activities.
However, there are some issues that should be addressed.
1. The figures in the manuscript are a bit blurry. Please replace them with clearer ones.
2. Acronyms that appear for the first time in the text should be given their full names first, for example , ITN in the part “Abstract”.
3. Some sentences may have grammatical errors which make them difficult to understand , for example, the phrase “predefined movement circuits” in line 60 should be checked for accuracy.
4. Please explain how these results of 0.65m and 0.76m in line 18 were calculated, as no such information is provided in subsequent sections.
Comments on the Quality of English LanguageThe manuscript need to be carefully examined and corrected before published.
Author Response
Reply to Reviewer #3
Dear reviewer,
Thank you very much for your time involved in reviewing the manuscript and your very encouraging and valuable comments on the merits. We have revised our manuscript accordingly.
To facilitate your review of this document, we will begin by retyping your comments in italic font, followed by our responses to those comments.
Comments :
Comments and Suggestions for Authors
This study evaluated the accuracy of a UWB tracking system (GenGee Insait KS) for tennis-specific movements by comparing it with an optical motion capture system (VICON). Ten amateur players (ITN 2–5) participated, performing seven exercises, including warm-up, agility drills, and tactical drills, with and without racquets. The UWB system demonstrated acceptable accuracy for steady movements but exhibited significant discrepancies during dynamic tennis-specific activities.
However, there are some issues that should be addressed.
Response :
Thank you for your considerable opinions on our revising procedure. Thanks for very much for your time and valuable feedback, which indeed help strengthen our manuscript. We have made revisions to our manuscript following the suggestions, and have highlighted all revisions in the manuscript using trace. The specific responses are listed below:
Comments 1:
The figures in the manuscript are a bit blurry. Please replace them with clearer ones.
Response 1:
The quality of the figures has been improved by replacing blurry images with higher-resolution versions to ensure they are clear and easily interpretable, which have been implemented in the updated manuscript.
Comments 2:
Acronyms that appear for the first time in the text should be given their full names first, for example, ITN in the part “Abstract”.
Response 2:
Thank you for your comment. We have ensured that all acronyms, including ITN in the Abstract, are introduced with their full names when they first appear in the text.
Comments 3:
Some sentences may have grammatical errors which make them difficult to understand, for example, the phrase “predefined movement circuits” in line 60 should be checked for accuracy.
Response 3:
Thank you for pointing that out. We have reviewed the sentences and revised them for clarity. The term "predefined movement circuits" has been replaced with a more accurate and clear expression: standardized movement protocols
Comments 4:
Please explain how these results of 0.65m and 0.76m in line 18 were calculated, as no such information is provided in subsequent sections.
Response 4:
Thank you for your observation. The 0.65m and 0.76m values mentioned in the abstract represent the average Root Mean Square Errors (RMSE) between the UWB and VICON systems for x and y coordinates, respectively. These values were derived from the box plots shown in Figure 2, where the dashed lines represent the average RMSE for both x and y coordinates. We have now clarified this in the revised manuscript to ensure the consistency of the results between the abstract and the figures.
Comments 5:
Comments on the Quality of English Language
The manuscript need to be carefully examined and corrected before published.
Response 5:
Thank you for your comment. The manuscript has been thoroughly reviewed and edited to improve the quality of English language throughout.
Round 2
Reviewer 1 Report
Comments and Suggestions for Authors
The authors have addressed all my concerns, I do not have further comments.
Reviewer 2 Report
Comments and Suggestions for Authors
I have carefully reviewed the authors’ responses to my comments and the corresponding revisions made to the manuscript. The authors have adequately addressed all the concerns raised in the first review, and the revised version reflects significant improvements in clarity, methodological rigor, and overall presentation.